# Epigenetic Mechanisms and Nephrotic Syndrome: A Systematic Review

**DOI:** 10.3390/biomedicines11020514

**Published:** 2023-02-10

**Authors:** Samantha Hayward, Kevon Parmesar, Gavin I. Welsh, Matthew Suderman, Moin A. Saleem

**Affiliations:** 1Translational Health Sciences, Bristol Medical School, University of Bristol, Bristol BS8 1UD, UK; 2MRC Integrative Epidemiology Unit, Population Health Sciences, Bristol Medical School, University of Bristol, Bristol BS8 1UD, UK

**Keywords:** nephrotic syndrome, FSGS, minimal change disease, SRNS, SSNS, epigenetics, micro-RNA, DNA methylation, long non-coding RNA, histone modification

## Abstract

A small subset of people with nephrotic syndrome (NS) have genetically driven disease. However, the disease mechanisms for the remaining majority are unknown. Epigenetic marks are reversible but stable regulators of gene expression with utility as biomarkers and therapeutic targets. We aimed to identify and assess all published human studies of epigenetic mechanisms in NS. PubMed (MEDLINE) and Embase were searched for original research articles examining any epigenetic mechanism in samples collected from people with steroid resistant NS, steroid sensitive NS, focal segmental glomerulosclerosis or minimal change disease. Study quality was assessed by using the Joanna Briggs Institute critical appraisal tools. Forty-nine studies met our inclusion criteria. The majority of these examined micro-RNAs (*n* = 35, 71%). Study quality was low, with only 23 deemed higher quality, and most of these included fewer than 100 patients and failed to validate findings in a second cohort. However, there were some promising concordant results between the studies; higher levels of serum miR-191 and miR-30c, and urinary miR-23b-3p and miR-30a-5p were observed in NS compared to controls. We have identified that the epigenome, particularly DNA methylation and histone modifications, has been understudied in NS. Large clinical studies, which utilise the latest high-throughput technologies and analytical pipelines, should focus on addressing this critical gap in the literature.

## 1. Introduction

Nephrotic syndrome (NS) is a clinical diagnosis comprised of a triad of high urinary protein levels, low blood albumin levels and fluid retention. NS can lead to end-stage kidney failure and a lifelong need for renal replacement therapy (dialysis or transplantation). NS is labelled as primary, when it occurs in isolation, or secondary when it occurs as the consequence of systemic disease, infection, or medication use. Primary NS can affect both children and adults, with a reported worldwide incidence of 2–7/100,000 people [1]. However, the clinical classification system used to subgroup primary NS differs between children and adults; children are stratified based on their initial response to high dose steroid treatment, whereas adults are grouped based on their kidney histology. This review focuses on primary NS which has been labelled as steroid resistant (SRNS), steroid sensitive (SSNS), minimal change disease (MCD) or focal segmental glomerulosclerosis (FSGS).

A breakthrough in our understanding of NS came through the investigation of hereditary NS, which identified causative genetic variants affecting podocyte (a key renal filtration cell) function. Therefore, we now understand that roughly 30% of patients with SRNS have genetically driven disease [2]. The exact disease mechanisms for other patients with NS remain elusive but are generally thought to be mediated by a variety of different immune mechanisms. T lymphocytes are believed to have a key role as some NS patients exhibit altered cytokine production compared with healthy controls [3,4]. B lymphocytes may be important in SSNS, as there is increasing evidence that these patients respond favourably to rituximab, a B cell-depleting treatment [5]. There is also convincing evidence that a subgroup of NS patients have disease caused by an imbalance of circulating factors. For example, plasma obtained from NS patients who exhibit disease recurrence after transplantation can induce aberrant expression of key slit diaphragm proteins in cultured human podocytes [6,7]. Therefore, it is likely that patients with nongenetic NS are not one homogenous group, but rather several distinct subgroups with different underlying pathogenic mechanisms, which are not yet fully understood.

Epigenetic mechanisms, such as DNA methylation (DNAm), micro-RNAs and histone modifications, alter gene expression without changing the underlying DNA sequence. These epigenetic mechanisms can be modified by a variety of environmental factors and can result in profound long-lasting changes in gene regulation. In conditions such as NS, where the disease cannot be solely explained by genetic variation, epigenetics may provide an answer. Epigenetic research is a rapidly expanding field that has contributed several biomarkers to clinical practice; for example, these biomarkers are being used to aid cancer diagnoses and predict response to treatment [8]. The aim of this systematic review is to identify, summarise and assess all published human research studies of epigenetic mechanisms in primary NS.

## 2. Materials and Methods

### 2.1. Protocol and Registration

The systematic review was designed using the Preferred Reporting Ideas for Systematic Review and Meta-analyses (PRISMA) systematic review checklist and was registered on PROSPERO, (ID: CRD42022311454, review protocol link: https://www.crd.york.ac.uk/prospero/display_record.php?RecordID=311454 (accessed on 17 February 2022)).

### 2.2. Search Strategy—Eligibility Criteria, Information Sources and Search Terms

Original research articles written in English and published before 17 February 2022 were eligible for inclusion. Studies that examined any epigenetic mechanism in samples collected from humans with SRNS, SSNS, FSGS, or MCD were included. Studies which solely included patients with membranous nephropathy or secondary NS were excluded. Studies that included patients with chronic kidney disease of varied aetiology, in which NS, SRNS, SSNS, FSGS, or MCD were not explicitly stated or the results for these specific diseases were not reported separately from other disease groups, were also excluded. Conference abstracts were excluded. Our inclusion criteria did not include any age restrictions as we wished to capture both adults and children with NS.

Studies were identified from two databases: PubMed (MEDLINE) and Embase. The search was performed by using the terms listed in Figure 1 and was last conducted on 17 February 2022.

### 2.3. Study Selection and Data Extraction

Duplicate articles were removed from the literature search results. The titles and abstracts of the remaining articles were screened against the eligibility criteria by two independent authors. Any discrepancies between the authors were identified and discussed (with input from a third author if required). The remaining included articles proceeded to full-text screening, using the same eligibility criteria, by two independent authors.

Data were extracted from the included studies by using a standardized data-extraction form created by the authors (Table 1). If the studies included work on both cell lines and patient samples, only data from the patient sample work were extracted. Only the results relating to SRNS, SSNS, FSGS, or MCD samples were extracted. If details of effect sizes were missing, the study was still included and available data extracted.

### 2.4. Critical Appraisal

Study quality and risk of bias was assessed by using the Joanna Briggs Institute (JBI) critical appraisal tools [9]. The risk of bias in the studies was categorised based on the percentage of “yes” scores in the JBI checklist: less than 50% was considered high risk of bias, 50–69% was considered moderate risk, and 70% or greater was considered low risk of bias.

All included articles were summarised, however in-depth reporting of results was limited to higher quality studies, defined as those at low risk of bias (JBI of 70% or greater) and which included ≥20 people with FSGS, MCD, SRNS, or SRNS.

## 3. Results

The search identified 708 articles, 219 from PubMed (MEDLINE) and 489 from Embase. Duplicate records (*n* = 166) and articles that did not meet the inclusion criteria on title and abstract screening (*n* = 483) were removed, resulting in 59 articles which proceeded to full-text screening. A further 10 articles were removed on full-text screening as they did not meet the inclusion criteria or did not have full texts available. In total, 49 studies were included in the review (Figure 2).

Forty-eight studies were case-control studies and one study had a repeated cross-sectional design. Micro-RNAs were the most studied epigenetic mechanism (*n* = 35, 71%) reported. Only five studies investigated DNAm (10%), four long noncoding RNAs (8%), two histone modifications (4%), two small RNA (4%), and a single study examined a circular RNA (2%). Blood was the commonly investigated tissue, but kidney and urine were also quite common (Figure 3). Twenty-four of the studies used samples that were collected exclusively from adults with NS and 13 used samples only from children. Seven studies included samples from both children and adults. Five studies did not explicitly state the participants’ ages at sample collection or give any inclusion or exclusion criteria based on age. Only 23 of the 49 studies met our higher quality criteria (results summarised in Table 2 and Table 3). The remaining low-quality studies are described in Appendix A.

### 3.1. Higher Quality Studies—Micro-RNAs

The higher quality micro-RNA studies varied in their approach with some opting to investigate specific candidate micro-RNAs (*n* = 10) and others utilising array-based technology (*n* = 9), which captures thousands of micro-RNAs (Table 2). Despite the different methodologies, there were corroborating findings between the studies. By using a micro-RNA array, Luo et al. demonstrated higher serum levels of miR-191 in children with NS compared to healthy controls [10]. Bayomy et al. showed higher serum levels of a micro-RNA from the same family, miR-191a-5p, in children with NS compared to controls using a candidate approach [11]. In NS adults, Ramezani et al. used an array to show increasing levels of serum miR-30c from healthy controls to people with FSGS and MCD, with the highest levels demonstrated in MCD patients [12]. Hejazian et al. adopted a candidate micro-RNA approach and also found increased levels of serum miR-30c-5p in NS patients [13]. In another study from the same authors, which utilised the same approach and possibly included some of the same patients, increased levels of serum miR-30c were observed in people with FSGS compared to healthy controls [14].

Comparable results were also observed in urine micro-RNA studies. Both Feng et al. and Chen et al. found higher urinary exosomal levels of miR-23b-3p and miR-30a-5p in children with NS compared to controls followed by a decrease in miR-23b-3p when patients were treated with steroids and achieved remission [15,16]. Feng examined only a small number of micro-RNAs, whereas Chen examined the whole transcriptome. Chen et al. also validated these findings in a second independent cohort. In addition, increased urine and serum miR-30a-5p were demonstrated in paediatric NS patients by Luo et al.; these levels also declined after steroid treatment and NS remission [16]. Zhang et al. identified higher urinary miR-30a-5p in adults with active FSGS, compared to remission [17]. However, in this disease setting, only patients who had steroid-responsive FSGS demonstrated a decrease in urinary miR-30a-5p after treatment.

### 3.2. Higher Quality Studies—DNA Methylation

Two studies examined DNAm, and both opted to investigate specific candidate regions (NLRP3 promoter; SOCS3 and SOCS5 promoters) and demonstrated differences in methylation between SRNS and SSNS patients (Table 3) [18,19]. The promoter region of NLRP3 was examined as hypomethylation of this region is known to affect gene expression and cause steroid resistance in acute lymphoblastic leukaemia [20]. Indeed, in NS lower DNAm of this region was demonstrated in steroid resistant patients compared to those who were steroid sensitive [18]. The SOCS3 and SOCS5 promoters were investigated as previous work by the authors had demonstrated increased plasma levels of these proteins in SRNS compared to SSNS and healthy controls [21]. In this study, the authors identified lower DNAm in the promoter region of SOCS3 in SRNS compared to SSNS [19].

### 3.3. Higher Quality Studies—Small RNAs

Small RNAs were investigated by two studies (Table 3); Duan et al. sought to explore whether the small RNA U6 varies across NS and other renal pathologies to determine its utility as an internal reference gene in micro-RNA studies. Williams et al. chose a whole transcriptome approach and demonstrated large numbers of differentially expressed small RNAs between FSGS and healthy controls, however, this study included only 48 patients and the findings were not validated in a second cohort.

## 4. Discussion

We have reported and summarized all published human studies of epigenetic mechanisms in NS. Overall, the epigenetics of NS is understudied, with only 23 high-quality studies published and 10 that attempted to replicate their findings in a second cohort of patients. Even in the higher-quality studies, the number of included patients were modest, with only seven studies including more than 100 patients. The research studies focused almost exclusively on micro-RNAs. Due to the high heterogeneity in methodology and the use of a diverse range of assays very few of the micro-RNA studies are truly comparable, allowing for only limited conclusions to be drawn. Despite this, concordant results were seen between a small number of the micro-RNA studies; serum miR-191, serum miR-30c, urinary miR-23b-3p, and urinary miR-30a-5p levels were observed to be increased in NS compared to healthy controls in multiple studies.

It is likely that epigenetic research in NS has been hampered by the fact that NS is a rare disease and so large numbers of patient samples are difficult to obtain. This will improve now that large national and multinational NS cohorts have been established, for example, the International Study of NS (International NephroS), the National Unified Renal Translational Research Enterprise (NURTuRE) and the NS Study Network (NEPTUNE) [22,23]. Comparability between studies should also improve due to technological and methodological advances in epigenetic research. The combination of high-throughput array technology, greater standardisation of analytic pipelines and a better understanding of patient characteristics that may confound analyses, should lead to more consistent approaches between research teams and hopefully, more reproducible results. Nephrologists can be inspired by other medical specialties, such as oncology, which have been quicker to invest in epigenetic research and are now reaping the rewards with successful translation of the results into clinical practice.

There are many promising clinical applications for epigenetic data given that epigenetic mechanisms are known to respond to and sometimes play key roles in biological responses to the environment and disease processes. The flexibility and reversibility of epigenetic states suggest that, in some cases, epigenetic mechanisms may be therapeutic targets. In fact, a few drugs which act as broad reprogrammers of the epigenome have entered clinical practice, such as the histone deacetylase inhibitor panobinostat for treatment of multiple myeloma [24]. More recently, the discovery that CRISPR-cas9 can be used to perform locus-specific epigenome editing will likely lead to targeted epigenetic therapies in the not-too-distant future [25]. However, epigenetic variation is useful to medicine beyond causal roles in disease development and progression. Any variation that is merely associated with environmental or genetic risk factors or to disease processes may be used as a biomarker to estimate disease risk, diagnose disease, predict disease progression, or predict treatment response. For example, in the United States, hypermethylation in the promoter regions of BMP3, NDRG4, SEPT9, and VIM genes have been approved for colorectal cancer screening [8]. Similarly, the methylation status of MGMT is widely used in glioma patients as a predictive biomarker of response to alkylating chemotherapy agents and is included on National Comprehensive Cancer Network guidelines [26]. Each of these biomarkers was discovered by comparing DNA methylation between case and control tissues. None is known to play a role in disease.

For any molecular mechanism to be successfully translated into a clinical biomarker, it must be obtained from an easily accessible tissue, demonstrate low interlaboratory variation in measurements and be sufficiently stable. The studies identified in this review examined epigenetic mechanisms in blood, urine, renal tissue, or a combination of these. Obtaining samples from any of these tissues would be acceptable in clinical practice, although the less invasive options of blood and urine would be preferential. In general, interlaboratory reproducibility is improving, particularly with the use of DNAm microarrays and the sharing of analytical methods [27]. However, differences in micro-RNA isolation protocols persist and can lead to biased measurements hindering clinical utility [28]. Finally, a benefit of epigenetic mechanisms is their stability, for example, the half-life of micro-RNAs is roughly 10 times longer than that of messenger RNAs, and changes in DNAm can persist throughout adulthood [29,30]. Interestingly, synthetic micro-RNAs, which are being developed as epigenetic drugs, are less stable than their endogenous counterparts and can be rapidly degraded and cleared from circulation, which is a key limitation [31].

## 5. Conclusions

Overall, the epigenome is an attractive field of research and in certain disease settings, epigenetic research is beginning to alter clinical practice. However, NS research in this area is lagging behind, with a lack of high-quality epigenetic research. In particular, DNAm and histone modifications have been woefully understudied. Established large NS patient cohorts, alongside the technological and methodological advances in epigenetic research, should allow this gap in the literature to be addressed in the near future.

**Table 2 biomedicines-11-00514-t002:** Summary of the higher quality micro-RNA studies.

Publication Details and Reference	Study Population	Epigenetic Data: Mechanism Studied and COVERAGE	Results: Key Findings	Repeated Epigenetic Measures	Replication	JBI Percentage and Risk of Bias
**Micro-RNAs—blood**						
Xiao, B et al. Cell Death & Disease, 2018. [32]	Aged 16–70. FSGS 102; IgAN 69; MPGN 24; Membranous 26; Healthy controls 129.	Micro-RNA—blood QuantoBio miRNA high-throughput assay—515 miRNAs (Discovery phase); Primer assays for miR-17, miR-451, miR-106a, miR-19b (Validation phase).	MiR-17, miR-451, miR-106a, and miR-19b were significantly downregulated in the plasma of FSGS patients compared with healthy controls, fold changes of 0.55, 0.56, 0.59 and 0.55 respectively (*p* < 0.05). A 4 miRNA (miR-17, miR-451, miR-106a, and miR-19b) FSGS classification model gave an AUC value of 0.82, *p* < 0.0001. A 3 miRNA (miR-17, miR-451, and miR-106a) FSGS remission classification model gave an AUC of 0.71, *p* < 0.01.	No	Yes	80% Low risk
Ardalan, M et al. PeerJ, 2020. [33]	Aged 20–60 FSGS 22; Membranous 30; Healthy controls 24.	Micro-RNA—PBMCs and plasma. MiRNA-135 primer assays.	Lower miR-135a-5p in patients with FSGS compared to controls, median relative expression 0.72 compared to 1.37, *p* = 0.046.	No	No	90% Low risk
Hejazian, S et al. International Journal of Nephrology and Renovascular Disease, 2020. [14]	Aged 20–60 FSGS 30; Membranous 30; Healthy controls 24.	Micro-RNA—PBMCs and plasma. MiR-30c and miR-186 primer assays.	Increased miR-30c level in PBMCs of patients with FSGS (0.004), compared to controls. Plasma miR-30c levels were not different between FSGS, Membranous or healthy controls.	No	No	90% Low risk
Rahbar Saadat, Y et al. Biofactors, 2020. [34]	Aged 20–60 FSGS 30; Membranous 30; Healthy controls 24	Micro-RNA—PBMCs MiR-24, miR-30a, and miR-370 primer assays.	Lower miR-24, higher miR-30a and higher miR-370 expression levels were observed in Membranous compared to FSGS (*p* = 0.040, *p* = 0.032, *p* = 0.041, respectively). There was no difference in the levels of these miRNAs between the control group and FSGS patients.	No	No	90% Low risk
Ni, F et al. Frontiers in Pediatrics, 2021. [35]	No specific age restrictions—only children included. Active NS before treatment 20; NS in remission 20; Healthy controls 20.	Micro-RNA—Blood—Th2 cells (CD4 + TCD25- cells) miR-24 and miR-27 primer assays.	Participants with active non-atopic NS had lower levels of miR-24 (mean 21.84 × 10^−3^) and miR-27 (mean 20.72 × 10^−3^) compared to healthy controls (46.03 × 10^−3^, *p* < 0.05, and 37.83 × 10^−3^, *p* < 0.05, respectively).	No	No	90% Low risk
Hejazian, S et al. Biofactors, 2020. [13]	Aged 20–60. NS 60; Healthy controls 24	Micro-RNA—PBMCs miR-30a, miR-30c, miR-186, miR-193, miR-217 primer assays.	Higher levels of miR-30c-5p (*p* = 0.005) and miR-193-3p (*p* = 0.041) were observed between NS patients and healthy controls. There was no difference in mi-RNA196-5p and miR-217 expression levels between NS patients and controls.	No	No	90% Low risk
Bayomy, N et al. Molecular Immunology, 2022. [11]	No specific age restrictions—only children included. SSNS 56; SRNS 24; Healthy controls 100.	Micro-RNA—blood miR-142-5p, miR-191a-5p, miR-181-5p, miR-30a-5p and miR-150a-5p primer assays.	NS patients had higher levels of the 5 studied microRNAs than healthy controls: miR-142-5p (mean expression 14.8 compared to 2.3, *p* = 7 × 10^−5^), miR-191a-5p (7.38 compared to 0.65, *p* < 10^−6^), miR-181-5p (2.25 compared to 0.41, *p* < 10^−6^), miR-30a-5p (1.49 compared to 0.72, *p* < 1 × 10^−6^) and miR-150a-5p (7.67 compared to 3.86, *p* <0.003). SRNS patients had higher levels of the 5 studied microRNAs than SSNS patients: miR-142-5p (mean expression 105.36 compared to 10.87, *p* < 10^−6^), miR-191a-5p (22.99 compared to 0.69, *p* < 10^−6^), miR-181-5p (4.22 compared to 1.39, *p* 10^−6^), miR-30a-5p (3.10 compared to 0.809, *p* < 10^−6^) and miR-150a-5p (10.97 compared to 6.26, *p* < 0.044). MiR-142a-5p had the best discrimination of NS patients from controls (AUC 0.965) and SRNS from SSNS (AUC 1.00) of a single mi-RNA.	No	No	90% Low risk
Zhang, C et al. American Journal of Kidney Diseases, 2015. [36]	Aged 18–69. FSGS nephrotic range proteinuria 78; FSGS remission 35; Membranous 63; DN 59; Healthy controls 69.	Micro-RNA—plasma TaqMan Low Density Array (Applied Biosystems)—384 human microRNAs.	Higher levels of miR-125b, miR-186, and miR-193a-3p were present in patients with FSGS relative to controls, (average fold changes of 5.77 *p* < 0.001, 3.04 *p* = 0.006, and 3.44 *p* < 0.001, respectively). MiR-125b and miR-186 concentrations were significantly lower in patients with FSGS in complete remission (*p* = 0.02 and *p* < 0.001) compared to those with nephrotic range proteinuria. In FSGS patients who achieved complete remission with steroid treatment, miR-125b and miR-186 levels declined markedly after they received steroids (SSNS, *p* = 0.002 and *p* = 0.002 respectively). There was no change in these miRNA levels in patients who did not respond to steroids (SRNS). Plasma miR-186, but not miR-125b, level was correlated with degree of proteinuria in patients with FSGS (R 0.185, *p* = 0.02).	No	Yes	100% Low risk
**Micro RNAs—urine**						
Altamemi I et al. Journal of Pharmaceutical Sciences and Research, 2019. [37]	No specific age restrictions—adults and children included. FSGS 24; Healthy controls 24.	Micro-RNA—urine miRNA-193a primer assay.	Urine miR-193a levels were higher in people with FSGS than controls (median fold change 2.125 compared to 0.375, *p* < 0.001). A fold change of >0.31 miR-193a gave an AUC of 0.829, sensitivity of 100% and specificity of 50% for identifying FSGS from controls patients.	No	No	80% Low risk
Zheng X et al. Experimental and Therapeutic Medicine, 2021. [38]	Aged 11–62FSGS 22; Membranous 36; NS—Nephropathy 22; Healthy controls 60.	Micro-RNA—urine miR-155 assay primers.	Urine miR-155 levels were higher in early NS than in control patients (roughly 4.5-fold compared to 1, *p* < 0.05). Urine miR-155 levels could distinguish early NS from control patients with an AUC of 0.9548, sensitivity of 93.27% and specificity of 92.58%.	No	No	90% Low risk
Wang N et al. Peer J, 2015. [39]	Aged 20–50 MCD 31; IgAN 120; Membranous 45; Healthy controls 40.	Micro-RNA—urine Affymetrix GeneChip miRNA 4.0 Array—2578 mature human miRNAs.	In the validation cohort the urinary level of miR-3613-3p were lower in IgAN compared with that in healthy controls, membranous and MCD, (relative levels: 0.27, 1.10, 0.58 and 0.61, respectively, all *p* < 0.01). There were no significant differences in miR-4668-5p between IgAN and patients with Membranous or MCD.	No	Yes	90% Low risk
Zhang W et al. Clinical Journal of The American Society of Nephrology, 2014. [17]	No specific age restrictions—adults included. Active FSGS 107; FSGS remission 103; Healthy controls 105; Membranous active 29; Membranous remission 26; DN with disease activity 23; Incipient DN 27; Validation: Healthy controls 27; FSGS remission 22; Active FSGS 33.	Micro-RNA—urine Taqman Low density Array—754 human miRNAs.	Urinary miR-196a, miR-30a-5p, and miR-490 discriminated patients with active FSGS from those in remission (AUC 0.92, 0.82 and 0.96 respectively). After steroid treatment, the levels of urinary miR-196a, miR-30a-5p, and miR-490 were lower in steroid-responsive FSGS patients (*p* < 0.001), but were unchanged in steroid-resistant FSGS patients. Urinary miR-30a-5p marginally predicted the response to steroid treatment in patients with active FSGS, (AUC 0.63, *p* = 0.03).	No	Yes	100% Low risk
Chen T, EBioMedicine, 2019. [16]	Aged ≤ 15 years. 126 NS Healthy controls 126.	Micro-RNA—urinary exosomes Illumina sequencing by synthesis—whole transcriptome.	The Illumina sequencing identified 30 urinary exosomal miRNAs which were increased in NS compared with controls (≥5 fold, *p* < 0.05), the top 15 proceeded to validation testing. Five mi-RNAs (miR-194-5p, miR-146b-5p, miR-378-3p, miR-23b-3p and miR-30a-5p) were significantly increased in NS in 2 independent cohorts (>3 fold, *p* < 0.01). During NS remission, all the mi-RNAs except miR-194-5p decreased, almost to the level of controls (*p* < 0.001).	Yes—before and after treatment	Yes	70% Low risk
Feng D, Translational Andrology and Urology, 2020. [15]	Aged ≤ 12 years NS active disease 68; NS remission 47; Healthy controls 50.	Micro-RNA—urinary exosomes miR-23b-3p, miR-30a-5p and miR-151-3p assay primers.	The levels of miR-23b-3p and miR-30a-5p decreased from the active NS group (10.58 and 18.57 respectively), to the remission NS group (8.63 and 15.62) to the controls (0.56 and 8.62, *p* all < 0.001). After treatment, levels of exosomal miR-23b-3p and miR-30a-5p in the active NS and remission NS groups decreased significantly (10.58 to 2.68 and 18.57 to 10.48; 8.63 to 2.43 and 15.62 to 9.63 respectively, *p* < 0.05). Urinary exosomal miR-23b-3p and miR-30a-5p could distinguish between children with NS and healthy children, (AUC 0.711, sensitivity 92.73% and specificity 59.09% for miR-23b-3p and AUC 0.844, sensitivity 85.71% and 63.64% for miR-30a-5p).	Yes—before and after treatment	No	100% Low risk
**Micro-RNAs—renal tissue**						
Yu J, et al. BMC Nephrology, 2019. [40]	No specific age restrictions—only adults included. MCD 4; FSGS 4; DN 4; Healthy controls 4. Validation: MCD 6; FSGS 6; DN 6; Healthy controls 6.	Micro-RNA—Renal—glomeruli and tubules miRCURY LNA Array, which contains 3100 miRNAs (covering all human, mouse and rat miRNAs).	Forty mi-RNAs were increased and 76 decreased in renal tissue from people with MCD, FSGS and DN compared with healthy donor kidney biopsy tissue. In a validation cohort, miR-3607-3p was decreased in people with MCD, FSSG and DN compared to healthy controls (*p* < 0.05). MiR-4709-3p was increased in people with MCD, FSSG and DN compared to healthy controls (*p* < 0.05).	No	Yes	80% Low risk
**Micro-RNAs—blood and urine**						
Ramezani A, et al. European Journal of Clinical Investigation, 2015. [12]	No specific age restrictions—adults and children included FSGS 16; MCD 5; healthy controls 5.	Micro-RNA—plasma and urine Affymetrix GeneChip miRNA 3.0 arrays—1733 human miRNAs.	Patients with FSGS had 126 differentially expressed miRNAs in plasma and 155 in urine compared to people with MCD. Plasma levels of miR-30b, miR-30c, miR-34b, miR-34c and miR-342 and urine levels of mir-1225-5p were higher MCD patients than in FSGS and healthy controls, (*p* < 0.001). Urinary levels of mir-1915 and miR-663 were lower in patients with FSGS compared to MCD and controls (*p* < 0.001). Urinary levels of miR-155 were higher in patients with FSGS compared to MCD and controls (*p* < 0.005).	No	No	90% Low risk
Zhang C, et al. Journal of Translational Medicine, 2018. [41]	Aged 16–65.FSGS with nephrotic range proteinuria 100; FSGS complete remission 100; Healthy controls 100; Replication cohort—FSGS 231.	Micro-RNA—plasma and urine TaqMan Human MicroRNA Array v3.0 A and B—754 human miRNAs.	Urinary miR-196a was significantly increased in active FSGS compared with FSGS patients in complete remission and healthy controls (*p* < 0.0001). There was no difference in plasma miR-196a between these patient groups. Urinary miR-196a was associated with proteinuria (*p* = 0.003), estimated glomerular filtration rate (*p* = 0.005), interstitial fibrosis (0.004), tubular atrophy (*p* = 0.38) and progression to end-stage kidney disease (<0.001). Multivariate Cox analysis confirmed urinary miR-196a as an independent risk factor for FSGS progression after adjusting for age, sex, proteinuria and eGFR (HR = 2.616, 95% CI 1.592–4.301, *p* < 0.001).	No	Yes	100% Low risk
Luo Y, et al. Clinical Chemistry, 2013. [10]	Aged 1–14 SRNS 24; SSNS 135; Healthy controls 109; Renal disease controls 44 (18 HSP, 15 IgAN, 11 lupus).	Micro-RNA—serum and urine TaqMan Low Density Array—754 human miRNAs.	Serum miR-30a-5p, miR-151-3p, miR-150, miR-191, and miR-19b were higher in NS children compared with healthy controls (median fold change 2.51, 2.56, 2.59, 3.29, 2.43 respectively, all *p* < 0.0001). Urinary miR-30a-5p was also increased in NS patients compared to healthy controls (median fold change 2.11, *p* = 0.001). The 5 serum mi-RNAs combined could distinguish NS from healthy controls, OR 40.7 (95% CI, 6.06–103; *p* < 0.0001). The concentrations of the 5 serum miRNAs and urinary miR-30a-5p declined after steroid treatment (*p* ≤ 0.002). All patients achieved remission with steroids.	Yes—before and after treatment	Yes	80% Low risk
Li J, et al. BioMed Research International 2017. [42]	Aged < 65 years FSGS 82; Membranous 84; DN 67; Healthy controls 72.	Micro-RNA—plasma and renal tissue MiR-217 assay primer.	There was no difference in miR-217 expression in renal tissue or plasma between healthy controls and people with FSGS.	No	No	90% Low risk

**Abbreviations:** Area under curve, AUC, diabetic nephropathy, DN; focal segmental glomerulosclerosis, FSGS; Henoch–Schonlein purpura (HSP); IgA nephropathy, IgAN; JBI, Joanna Briggs Institute; membranoproliferative glomerulonephritis, MPGN; minimal change disease, MCD; nephrotic syndrome, NS; peripheral blood mononuclear cells PBMCs; steroid-resistant NS, SRNS; steroid-sensitive NS, SSNS.

**Table 3 biomedicines-11-00514-t003:** Summary of the higher quality DNA methylation and small RNA studies.

Publication Details and Reference	Study Population	Epigenetic Data: Mechanism Studied and Coverage	Results: Key Findings	Repeated Epigenetic Measures	Replication	JBI Percentage And Risk Of Bias
**DNA methylation**						
Locafo M, et al. Clinical and translational science, 2021. [18]	No specific age restrictions—adults and children included. Adults: SRNS and FSGS 10; SSNS 18 (of which, 13 MCD and 5 FSGS); Validation: Children: FSGS 2; MCD 14.	DNAm—PBMCs Bisulphite conversion and NLRP3 promoter primer assay.	In both adults and paediatric patients, NLRP3 promoter methylation was significantly reduced in SRNS compared with SSNS (median 0.2, and 0.33 respectively, *p* = 0.00024). NLRP3 methylation distinguished between SRNS and SSNS with AUC 0.736 in children (*p* = 0.00097) and 0.867 in adults (0.00019).	No	Yes	90% Low risk
Zaorska K, et al. Acta Biochimica Polonica, 2016. [19]	No specific age restrictions—children included. SRNS 40; SSNS 36; Healthy controls 33.	DNAm—blood. Methylation primers (methylation-specific PCR)—CpG islands for SOCS3 and SOCS5.	There was lower DNAm at one region of SOCS3 promoter in SRNS compared to SSNS, 82.5% of SRNS patients were unmethylated in this region compared to 17.7% of SSNS patients (*p* < 0.0001).	No	No	100% Low risk
**Small RNAs**						
Duan Z, et al. Scientific Reports, 2018. [43]	Aged > 18. FSGS 25; MCD 37; IgAN 330; Healthy controls 130; Membranous 90; HSP 5; Non-IgA MPGN 5; Renal amyloidosis 2.	Small nuclear RNA—urine. U6 primer assay.	No significant difference in the expression of U6 between the patients with IgAN, Membranous, MCD and FSGS.	Yes—before and after treatment in IgAN.	Yes	100% Low risk
Williams A, et al. Kidney international, 2022. [44]	No specific age restrictions—adults included. FSGS 38; Healthy controls 10	Small RNA—renal—glomeruli and tubules Illumina TruSeq small RNA library preparation kit—whole transcriptome.	Large numbers of small RNAs, including microRNAs, 3′-transfer RNA fragments, 5′- transfer RNA fragments, and mitochondrial transfer RNA fragments, were differentially expressed between histologically indistinguishable tissue regions from patients with FSGS and controls. In FSGS, miR-21-5p progressively increased and miR-192-5p progressively decreased in glomerular and tubulointerstitial regions with increasing levels of histological damage.	No	No	100% Low risk

**Abbreviations:** Area under curve, AUC; DNA methylation, DNAm; focal segmental glomerulosclerosis FSGS; Henoch–Schonlein purpura HSP; IgA nephropathy, IgAN; JBI, Joanna Briggs Institute; membranoproliferative glomerulonephritis MPGN; minimal change disease, MCD; nephrotic syndrome, NS; peripheral blood mononuclear cells, PBMCs; steroid-resistant NS, SRNS; steroid-sensitive NS, SSNS.

## Figures and Tables

**Figure 1 biomedicines-11-00514-f001:**
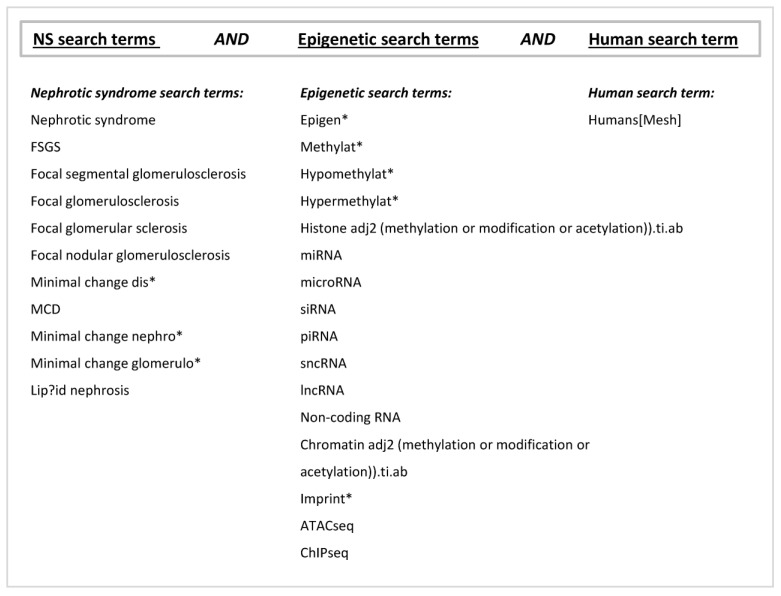
Systematic review search terms. Search term symbols: * trunation (broadens the search to include any ending to the word); wildcard character (in the search the character can be substituted for zero or one character of a word).

**Figure 2 biomedicines-11-00514-f002:**
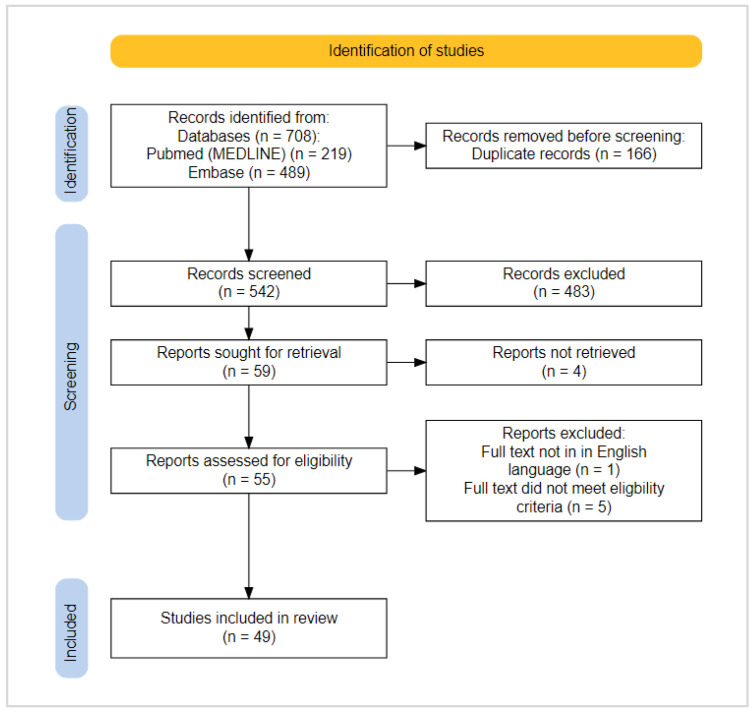
PRISMA flowchart.

**Figure 3 biomedicines-11-00514-f003:**
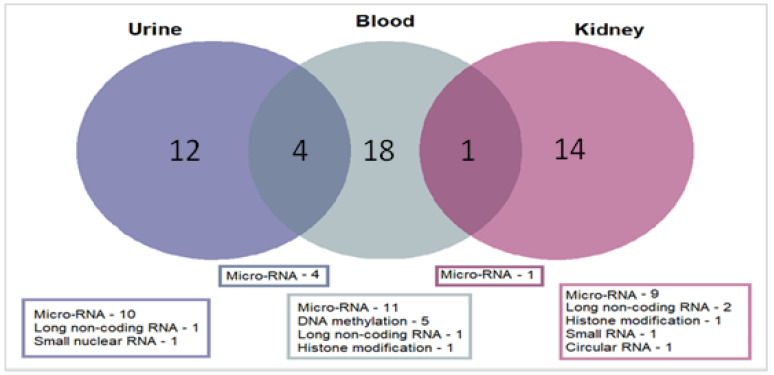
Number of studies by epigenetic mechanism and tissue.

**Table 1 biomedicines-11-00514-t001:** Systematic review data extraction form.

	Data	Comments
**Study Design**		E.g., Case-control, cohort study etc.
**Study population**	Sample size	Total number of participants and number with NS
	Diagnosis and control group	E.g., SRNS v age matched controls
	Age	E.g., 0–18 years only
**Epigenetic data**	Mechanism studied	E.g., DNA methylation
	Tissue studied	E.g., Lymphocytes
	Data generation approach and coverage	E.g., 450 k Illumina array
**Research objectives**		Directly as stated in the article
**Analysis**	Exposure	E.g., The epigenetic mechanism
	Outcome	E.g., Treatment response, disease subgroup
	Confounders	E.g., Age, sex, cell type proportions
	Mediation analysis	Yes/No and details if Yes
	Mendelian randomization	Yes/No and details if Yes
	Machine learning	Yes/No and details if Yes
	Other ‘omics’ data incorporated	Yes/No and details if Yes
	Repeated measures of epigenetic data	Yes/No and details if Yes
	Replication or validation attempted	Yes/No and details if Yes
**Results**		Key findings (state effect size and statistics)

## Data Availability

This study did not involve any newly generated data.

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
