# Peer review of "Epigenetic Mechanisms and Nephrotic Syndrome: A Systematic Review"

_biomedicines, 2023, doi:10.3390/biomedicines11020514_

Round 1
Reviewer 1 Report
Review of biomedicines-2190329
Epigenetic mechanisms and Nephrotic syndrome: a systematic review
Samantha Hayward, Kevon Parmesar, Gavin I Welsh, Matthew Suderman and Moin A Saleem
General comments
This manuscript describes the published human studies of epigenetic mechanisms in nephrotic syndrome. The topic addressed is interesting. I have annotated the manuscript with several minor corrections, which I believe will improve the readability of the paper.
Specific comments
1, Do the authors think that using the methodology words such as “ATACseq” “ChIPseq” for the search term increase the literature?
2, In Figure1, “Lipi?id” might be typo.
Author Response
Reviewer 1:
This manuscript describes the published human studies of epigenetic mechanisms in nephrotic syndrome. The topic addressed is interesting. I have annotated the manuscript with several minor corrections, which I believe will improve the readability of the paper. Specific comments:
- 1, Do the authors think that using the methodology words such as “ATACseq” “ChIPseq” for the search term increase the literature?
Thank you for this helpful suggestion. We have improved the manuscript by adding in these two additional search terms and re-running the search, (with the date limit set to match the date the original search was carried out). Three additional papers were identified in the updated search; the abstracts were screened by two reviewers and were excluded as they did not fulfil the inclusion criteria (one was a review paper and two did not include patients with nephrotic syndrome). The text and figures within the manuscript have been edited to reflect these changes in the search and article numbers.
- 2, In Figure1, “Lipi?id” might be typo.
The search term which was run was ‘Lip?id’ and figure 1 has been updated accordingly. The ‘?’ is an optional wild card character within the search which can stand for zero or one character within the word. Therefore this search term would capture both ‘Lipoid nephrosis’ and ‘Lipid nephrosis’.
Reviewer 2 Report
This systematic review is to identify, summarise and assess all published human research studies of epigenetic mechanisms in primary nephrotic syndrome. How the author distinguished the higher quality studies vs. lower quality studies? What is the standard? It is crucial that a review article discusses the available information in a non-biased way (even if this does not support the author's "desired" hypothesis).
Author Response
Reviewer 2:
This systematic review is to identify, summarise and assess all published human research studies of epigenetic mechanisms in primary nephrotic syndrome. How the author distinguished the higher quality studies vs. lower quality studies? What is the standard? It is crucial that a review article discusses the available information in a non-biased way (even if this does not support the author's "desired" hypothesis).
Thank you for highlighting that how we assessed study quality was not clear. The Joanna Briggs Institute (JBI) Critical appraisal tools were used to assess the risk of bias in each study and patients with a score of under 7 were excluded. The JBI tools are widely used to assess risk of bias and study quality when conducting systematic reviews.
We have updated the manuscript to make our approach clearer and to aid the interpretibility of the JBI scores. We have transformed the JBI scores to percentages and categorised these scores in to ‘high’, ‘moderate’ or ‘low’ risk of bias. We have also included an extra column in all the results tables which gives the JBI percentage score and category. The JBI category classification thresholds which we chose have been widely used in the scientific literature, (for example, Sampaio et al, https://doi.org/10.1016/j.rmed.2019.03.018; Weber Mello et al, https://doi.org/10.1111/jop.12726 and Goplen et al https://doi.org/10.1186/s12891-019-2619-8). We believe that these changes have strengthened the manuscript.
Round 2
Reviewer 2 Report
I have no further comments